# Impact of Unsustainable Environmental Conditions Due to Vehicular Emissions on Associated Lifetime Cancer Risk in India: A Novel Approach

**DOI:** 10.3390/ijerph19116459

**Published:** 2022-05-26

**Authors:** Parteek Singh Thind, Deepak Kumar, Sandeep Singh, Jasgurpreet Singh Chohan, Raman Kumar, Shubham Sharma, Changhe Li, Gianpaolo Di Bona, Antonio Forcina, Luca Silvestri

**Affiliations:** 1Department of Civil Engineering, Punjab Engineering College, Chandigarh 160012, India; prateek.thind@gmail.com; 2Department of Geography, Kurukshetra University, Kurukshetra 136119, India; deepak.geo.earth@gmail.com; 3Department of Civil Engineering, University Centre for Research & Development, Chandigarh University, Mohali 140413, India; drsandeep1786@gmail.com; 4Department of Mechanical Engineering, University Centre for Research & Development, Chandigarh University, Mohali 140413, India; jaskhera@gmail.com (J.S.C.); ramankakkar@gmail.com (R.K.); 5Department of Mechanical Engineering, IK Gujral Punjab Technical University, Main Campus, Kapurthala 144603, India; 6School of Mechanical and Automotive Engineering, Qingdao University of Technology, Qingdao 266520, China; sy_lichanghe@163.com; 7Department of Civil and Industrial Engineering, University of Cassino and Southern Lazio, 03043 Cassino, Italy; 8Department of Engineering, University of Naples “Parthenope”, 80133 Naples, Italy; antonio.forcina@uniparthenope.it; 9Department of Engineering, University of Rome “Niccolo Cusano”, 00166 Rome, Italy; luca.silvestri@unicusano.it

**Keywords:** environmental conditions, elemental carbon, cancer risk assessment, Indian Western Himalayas, sustainable development

## Abstract

The Indian Western Himalayas (IWHs) are a world famous tourist spot, and every year millions of tourists visit this area in fossil fuel-driven vehicles. Emissions from these vehicles persistently deteriorate the pristine environment of the IWHs. Therefore, in the current study, efforts were made to assess the compromised environmental conditions of Manali, Himachal Pradesh, India that resulted from the inflow of tourists and the activities undertaken by them. This study revealed that Manali could sustainably accommodate only 0.305 M tourists/month, and this threshold was reported to be crossed in the months of April, May and June. Furthermore, to augment these findings, water and ambient air samples were collected and analyzed for the presence of elemental carbon (EC) from one of the medium tourism potential regions of Manali, i.e., the Hamta glacier. The tributary receiving water from the Hamta glacier and the ambient air of the area was observed to be contaminated with 42 ± 12 ppb and 880 ± 43 µg m^−3^ of EC, respectively. It was observed that the inhalation and ingestion of EC-contaminated air and water could jeopardize human health due to a high lifetime cancer risk. However, without the intervention of eco-tourism in the study area, higher environmental health effects were also speculated. The observations made in this study are expected to trigger the interests of the researchers, international scientific community and regional authorities working towards the unsustainable development of the IWHs and deteriorating environmental conditions.

## 1. Introduction

Comprising 22% of the global land mass, mountainous terrains are home to 13% of the global population, out of which ~16% reside more than 2500 m above the mean sea level (msl) [1,2,3,4,5,6,7]. Glaciers, located in these mountainous terrains, act as a fresh water resource and fulfill the water-related requirements of about half of the global population. However, these fresh water reservoirs are undergoing rapid changes due to human interference-induced climate change, infrastructure development, industrialization, and tourism. The increased in-flow of tourists, in glaciated areas, is usually complemented by the intensified biomass burning, transportation and vehicular density. Fossil fuels’-driven vehicles emit several toxic compounds such as volatile organic compounds (VOCs), heavy metals, poly aromatic hydrocarbons (PAHs), elemental carbon (EC) etc., ref. [1]. In this way, vehicular emissions contaminate the different environmental media of the glaciated terrains [8,9,10,11,12,13,14].

EC, among other toxicants, is also resistant to several atmospheric degradation pathways i.e., photo-induced, chemical and biological degradation. Hence, it is susceptible to long range atmospheric transport and can even travel miles away from the source of emission [15,16,17,18,19,20,21,22,23,24,25,26,27]. EC has a cancer potency factor equivalent to carcinogenic compounds such as benzo[a]pyrene, and upon being bio-accumulated in the environment it can affect human health [26]. EC, compared to other particulates, it has smaller size and can penetrate through the walls of the respiratory organs and enter into the blood stream of the body and reach other organs [8,21,22,26,27,28,29,30,31,32,33,34,35,36,37,38,39,40]. Moreover, EC can disturb the autonomic nervous system and, subsequently, increase the susceptibility of the heart to fatal dysrhythmias [3,9,10,21,22]. EC, emitted from diesel-driven vehicles, has also been reported to induce lung cancer and blood cancer in human sand, therefore, people in occupations involving frequent transportation and travelling are more susceptible to such cancers [14,23,24,25]. In a meta-analysis performed by Cohen and Higgins (1995), it was revealed that exhaust from a diesel engine comprising EC has a similar relative risk as that of cigarette smoke [6]. However, the issue of the compromised health of the residents of the Indian Western Himalayas (IWHs), due to increased tourism, vehicular density and contamination of environment with EC, still needs to be addressed.

The IWHs is a landlocked area which receives emissions from urban regions of India such as Delhi, Raipur, Gwalior and Lucknow. These urban areas of India are listed among the world’s top ten most polluted cities [26]. The Indo-Gangetic plains (IGPs) are also one of the neighboring areas of the IWHs. Several sources of emission such as stubble burning, forest fires, etc., are located in the IGPs and hence were also reported to significantly contribute in the atmospheric concentration of EC in the IWHs [20,32,41]. Currently, the tough terrain of the IWHs are also connected with the metropolitan cities of the India because of increased tourism and economic activity [12]. The increase in tourists also contaminates the pristine surroundings of the mountainous landscapes through activities such as, fossil fuel combustion, domestic-biomass burning, vehicular emissions, etc. [11,13,28] In this way, the atmospheric conditions of the IWHs also get contaminated with a high concentration of EC [31]. A study conducted at Nagpur showed that a significant amount of EC was present in snow samples of this region [18,19].

A scientific analysis has been performed to identify the scope of various studies carried out in the area of cancer due to excessive vehicular emission. The keyword analysis chart (Figure 1) indicates that significant work has been performed in the area of exhaust gas, occupational exposure and lung diseases. However, lesser work has been done on the domains of air composition and cancer risk assessment in tourist places. Limited research work has been done on Indian tourist cities regarding cancer risk and vehicular emissions. In the harsh weather conditions of the IWHs, tourism is the most promising way for the residents to earn their livelihood. However, due to the uncontrolled inflow of tourists in this region, drastic impacts on the ecosystem and environment were observed [30].

Therefore, the current study focuses on estimating the tourism carrying capacity (TCC) of one of the world-famous tourist spots of the IWHs, i.e., Manali, Himachal Pradesh, India. Furthermore, to compliment the findings of the study from a health point of view, various samples of different environmental media, i.e., water and ambient air, were collected near the snout of the Hamta glacier which is located near Manali. The lifetime carcinogenic and non-carcinogenic risk assessment of the residents exposed to the contaminated environment of the study area was also estimated. The findings of this study may provide the required impetus in framing the required policies to encourage eco-tourism so that the health disorders linked with human exposure to toxic emissions can be minimized.

## 2. Study Area

### 2.1. Tributary of Hamta Glacier

In 10 June 2017 water samples were collected from the stream fed by the melting of the Hamta glacier. The ambient air concentration of EC was measured using a hand-held microAeth^®^ AE51 (AethLabs, San Francisco, CA, USA). The location of the sample collection sites is shown in Figure 2.

### 2.2. Manali, Himachal Pradesh

For TCC based analysis, Manali, Himachal Pradesh, India was chosen as the study area. The geographical location of Manali is shown in Figure 3. The Manali terrain, often termed as the Crown of Himachal Pradesh, has a distinct geographical location and its coordinates are 32.2396° N and 77.1887° E. Major tourist destinations of Manali such as the Solang Valley, Rohtang Pass, Hamta trek, Jagatsukh trek, Beas Kund etc. are covered with snow, and these places have an inflow of approximately 110,000 tourists per year [12]. The flourishing tourism industry of Manali demands a reduction in the use of fossil fuel reliant vehicles, which act as a potential source of EC, and should instead promote the use electronic vehicles.

## 3. Materials and Methods

A brief outline of the methodology adopted for the current study is shown in Figure 4.

### 3.1. Field Sampling and Laboratory Analysis

Triplicates of the water samples were collected in high-density polyethylene (HDPE) bottles from the tributary of the Hamta glacier. These samples were brought to the campsite for preliminary analysis. In addition, water samples were filtered through quartz filter paper (Whatman^®^ QMA; pore size = 1 µm) using a vacuum filter assembly, as also adopted by Wang et al. (2012) [39]. Subsequently, to remove inferences in the chemical analysis, the quartz filter papers were first pre-heated at 800 °C for 24 h. They were then stored in a desiccator which was brought along to the camp site in a sealed container. In order to minimize the loss because of the adherence of particles to the walls of the containers, water samples were sonicated for 15 min.

After the filtration process, the filter papers were treated with HCl (1–3% *v*/*w*) and kept inside the petri-slides (Millipore^®^). These petri-slides were packed and then transferred to the Institute’s laboratory for further analysis. In the laboratory, estimation of EC on the filters was done using a carbon analyzer (Atmoslytic Inc., Calabass, CA, USA). Blank samples were also prepared and analysed using a similar procedure.

While performing ambient air monitoring, an AE51 was located at a height of ~1.5 m above the ground level and near the snout of the Hamta glacier to monitor the ambient air concentration of EC for 8 h/d. The AE51 with full battery was kept at the monitoring location from 8:00 a.m. to 4:00 p.m., from 6 June 2017 to 15 June 2017. Each day, the old filter ticket was removed and replaced with a new one. All of the data readings stored in the equipment were later entered into the laptop. The operating conditions of the instrument are described in Table 1.

Attenuation of light (ATN) caused by loaded aerosols is calculated by Equation (1).
(1)$$\mathrm{ATN}=100\times \ln\left(\frac{I_{o}}{I}\right)$$
where, I_o_ is the intensity of light through the fibrous filter having zero deposition and I is the intensity of light passing through used filter. Further, *EC* concentration is calculated using Equation (2).
(2)$$EC_{conc}=\frac{10^{9}}{\sigma_{ATN}}\times \left[\frac{A\times \Delta ATN}{100\times Q\times \Delta t}\right]$$
where, *EC_conc_* is the concentration of EC in ng m^−3^; *A* is the area of the aerosol spot on the filter in m^2^; *Q* is the flow rate of the pump in m^3^ s^−1^; Δ*t* is the duration of monitoring in s; Δ*ATN* is the change in the *ATN* value during the monitoring of the ambient air, $\sigma_{ATN}$ is the apparent mass attenuation cross-section for the EC in m^2^ g^−1^.

### 3.2. Identification of Tourism Potential Regions

Tourism potential (TP) regions were identified on the basis of the availability of different factors of tourists’ attraction such as, natural resources, cultural resources, mountain sports facilities and infrastructural facilities [2,5]. Using a modified multi-criterion evaluation (MCE) approach, TP regions were identified and categorized as high, medium and low TP regions. Indicators and sub-indicators were assigned weightages and ranks merely on the basis of knowledge, as mentioned in Equation (3).
(3)$$\begin{array}{ll} \mathrm{TP}=\mathrm{NW} & \overset{6}{\underset{N=1}{\sum }}\mathrm{Nw}\\ & +\;\mathrm{CW}\overset{4}{\underset{C=1}{\sum }}\mathrm{Cw}+\mathrm{AW}\overset{6}{\underset{A=1}{\sum }}\mathrm{Aw}+\;\mathrm{FW}\overset{5}{\underset{F=1}{\sum }}\mathrm{Fw} \end{array}$$
where, TP depicts the tourism potential of the study area; NW, CW, AW and FW are the normalized weights for indicators such as, natural resources (0.25), cultural resources (0.12), adventure and sports (0.25) and facilities and infrastructure (0.38), respectively. Similarly, NW, CW, AW and FW present the normalized weights for their sub-indicators and their values are mentioned in Table 2.

### 3.3. Tourism Carrying Capacity

For Manali, TCC was evaluated using Equations (4) and (5):(4)$$\mathrm{PCC}=\left(\frac{A}{\mathrm{Au}}\right)\times \;\mathrm{Rf}$$
where PCC is the physical carrying capacity; A signifies the availability of recreational area; Au is the minimum recreational area required for the comfortable stay of a single tourist and Rf depicts the rotation factor. Rf is calculated by dividing the open tourist hours with average duration of stay of a tourist in Manali. Similarly, A was calculated using the similar approach as used in NEERI (2012) and information acquired from the tourism department of Manali [18]. Minimum recreational area i.e., Au, was estimated using the findings of Visitor Experience and Resource Protection (VERP) framework [36] (US Department of the Interior, 1997) and modified People at one time (PAOT) criteria [4,15]. According to the preliminary survey, Rf was estimated to be 1 for Manali. Furthermore, tourism carrying capacity (TCC) was calculated using Equation (5).
(5)$$\mathrm{TCC}=\mathrm{PCC}\;\times \;({\mathrm{cf}}_{1}\;\times \;{\mathrm{cf}}_{2}\;\times \;{\mathrm{cf}}_{3}\;\ldots \ldots \ldots \ldots \ldots ..\times \;{\mathrm{cf}}_{n})\;$$
where, cf_1_ to cf_n_ denotes the correction factors for different variables used for the calculation of ERCC. Correction factors were calculated using Equation (6).
(6)$${\;\mathrm{cf}}_{v}=1-(L_{\mathrm{mv}}/T_{\mathrm{mv}})$$
where, cf_v_, Lm_v_ and Tm_v_ denote the correction factor, limiting magnitude and the total magnitude of variable v, respectively. Tourism in Manali is dependent on numerous environmental and social factors, therefore it is also relevant to identify these factors. According to the preliminary survey, we identified that in Manali there were six environmental and social factors: temperature, infrastructure, rainfall, management, perception and transport, which affected tourism.

### 3.4. Health Risk Assessment

Human exposure to EC can induce carcinogenic health impacts. Human beings are exposed to EC via three major pathways i.e., ingestion, inhalation (both oral) and by physical contact (dermal). The cancer potency factor for EC has not yet been estimated, therefore studies emphasizing the impacts of EC on human health are limited. In this study, as ECis largely comprised of Benzo-a-pyrene (B[a]P) [26], the latter’s slope factor was used to calculate its carcinogenic health impacts.

#### 3.4.1. Health Risk through Oral Intake

Human health risks associated with ingestion of EC-contaminated water and inhalation of EC-contaminated air can be calculated using Equations (7) and (8).
(7)$$Intake\;\left(oral\right)=\frac{C\times IR_{i,j}\times EF_{i,j}\times ED_{i,j}}{BW\times AT}$$
(8)$$\mathrm{Life}\;\mathrm{time}\;\mathrm{cancer}\;\mathrm{risk}=Intake{\;}_{oral}\times SF$$
where, *Intake (oral)* is the amount of EC (as a proxy of B[a]P) being consumed i.e., ingested and inhaled, by human beings in a single day (mg kg^−1^ d^−1^); C is the concentration of EC reported in the water and atmosphere of the glacier (mg L^−1^). A description of each of the parameters used during the calculation of intake (oral), is mentioned in Table 3.

The intake rate (oral) calculated using the parameters were further used to calculate the lifetime cancer risk, as mentioned in Equation (8). The lifetime cancer risk from oral exposure of EC was calculated by multiplying the intake rate (oral) with its slope factor, as mentioned in Equation (9) and Table 4.
(9)$$\mathrm{Life}\;\mathrm{time}\;\mathrm{cancer}\;\mathrm{risk}=Intake{\;}_{Exp}\times SF$$
where *Exp* denotes the type of exposure i.e., oral or dermal, and *SF* is the slope factor of EC (as a proxy of B[a]p). SF for EC is shown in Table 4.

#### 3.4.2. Health Risk through Dermal Intake

Glacier-fed water is used for several domestic purposes, and in this way, the skin of human beings may come into contact with the EC-contaminated water. Therefore, the lifetime cancer risk associated with exposure of humans to EC via dermal contact was also evaluated using Equation (10).
(10)$$Intake\;\left(dermal\right)=\frac{DA\times EV\times SA\times EF\times ED}{BW\times AT}$$
where, *intake (dermal)* is the amount of EC with which human beings are exposed via dermal contact in a day (mg kg^−1^ d^−1^). Similarly, event frequency i.e., *EV*, represents the number of times a human being is exposed to EC in a single day. The value of *EV* was assumed to be 1 for both adults and children. *DA* is the dermal absorption and it shows the amount of EC with which a unit area of human skin is exposed to (mg cm^−2^). *SA* is the area of skin and it depicts the total area of human skin which is exposed to EC during a single event. *DA* and *SA* are evaluated using Equations (11) and (12)
(11)$$DA\;(\mathrm{mg}/{\mathrm{cm}}^{2})=K\times C\times t\times CF$$
(12)$$SA\;\left({\mathrm{cm}}^{2}\right)=239\times H^{0.416}\times BW^{0.517}$$

Description of the various parameters used for estimating *DA* and *SA* are described in Table 5.

## 4. Results and Discussion

Lifetime cancer risk from an EC-contaminated atmosphere was estimated for adults and children. The results did not depict any significant effect, however, with the current rate of EC emissions. Regardless, a future carcinogenic impact can be assumed.

### 4.1. Tourism Potential Regions

The identified tourism potential regions in Manali are categorized into three regions i.e., high TP region, medium TP region and low TP region on the basis of the indicators described earlier in the article.

**High TP region:** This consists of the most famous tourist spots in the Manali. This region comprises Manali city (i.e., Mall Road area), Rohtang Pass and the Solang Valley. These spots attract the largest number of domestic and foreign tourists in the Manali. This region fulfils most of the basic needs of the tourists due to which increases its popularity. Along with infrastructure and transportation facilities this region is also rich in social and cultural amenities and famous tourist spots such as Mall Road, the Sports Complex, the Hadimba Temple and water sports, etc., Rohtang Pass is guarded by snow covered mountains. It is also famous as it is world’s highest road that can accommodate motorized vehicles, adding adventure to the excitement of the tourists. The calm, relaxing and leisurely lifestyle in this region adds a true spirit to hill station. Solang Valley is very famous among tourists for its adventure sports such as horse riding, skiing and.

**Medium TP region:** The areas in this region are suitable for hiking and also have religious places located at high altitudes, where physical strength is required to reach them. According to the present study, Medium TP regions are the Jogni Falls, Beas Kund and Hamtah Pass. These places are endowed with natural beauty. They are mainly visited by professionals. Foreign tourists are attracted because of the adventure and isolation and other facilities that it offers.

**Low TP region:** Due to lack of infrastructure and lack of accessibility, very few tourists are attracted to this region. High altitude tourist spots, valleys, glaciated areas and hiking treks are the destinations that fall in this region. Some of the important spots of this region are Hanuman Tibba trek, Deo Tibba trek, Jagatsukh Trek, and the Jagatsukh Glacier, etc. In such tourist spots the main tourist attractions are snow, glaciers, snow covered peaks, and camp fires in the wild.

### 4.2. Estimating the Tourism Carrying Capacity

The TCC was evaluated to find out the highest number of tourists that Manali can handle. The values of the different variables required for the estimation of the TCC of Manali are shown in Table 6.

According to Table 6, it was inferred that the TCC for Manali is 300,000 tourists/month. However, it was reported that tourist flow varies significantly throughout the whole of the year, and therefore in this study a comparison of monthly tourist flow for the year 2017 in Manali with its TCC was done, and the results are shown in Figure 5.

Figure 5 shows that the number of tourists visiting Manali is below the TCC throughout the year except for a few months. Furthermore, it can also be seen that during the months of April, May, June and October, the flow of tourists surpassed the TCC of Manali. During these months, various urban management problems such as traffic jams, the unavailability of rooms, the littering of solid waste etc., are faced by the local authorities. In the trekking spots of the low TP regions, large amounts of deforestation resulting from lodging and camp fires was reported. The unmanageable generation of solid waste and wastewater from hotels was also a matter of concern.

### 4.3. Carcinogenic Health Impacts of EC

The concentration of EC in different environmental media is mentioned in Table 7.

From Table 7, it is evident that the water body receiving water from Hamta glacier was reported to be contaminated with 19.02 ppb to 54 ppb of EC. Water from these contaminated streams was used by human beings for drinking and bathing, thus exposing them to EC. Other than that, the atmospheric EC was observed to be contaminated with 880 ± 43 µg/m^3^ of the EC. This contaminated air, when inhaled by human beings. Can become deposited in the lungs, thus making them susceptible to health issues. Hence, human beings get exposed to EC via three pathways i.e., ingestion, inhalation and dermal absorption. In present study, life time cancer risk assessment was investigated to estimate the carcinogenic impacts of EC on humans (adults and children). The results obtained via this investigation are shown in Table 8.

It is evident from Table 8 that EC, at its present deposition rate, did not pose carcinogenic health impacts on adults. However, the lifetime cancer risk for children was reported to be <10^−6^ in cases of ingestion and inhalation. Therefore, children are prone to carcinogenic and health effects from being exposed to a contaminated environment. It has also been reported that in the previous few years the number of cancer patients was also on rise, and such a trend can be associated with the contamination of the environment with carcinogenic compounds [12]. It should be noted that when the value of the lifetime cancer risk is less than 10^–6^, the carcinogenic impacts can be speculated. However, when the value is closer or between 10^–4^ and 10^–6^, non-carcinogenic health impacts may be the only ones that occur. Therefore, with the increasing air pollution and emission of EC, the value of the lifetime cancer risk can be presumed to increase. In this way, an EC-contaminated environment may affect human health in mountainous regions.

If strict measures are not adopted to prevent vehicle emissions, the residents of the area under study are more susceptible to serious health issues. Hence, it is recommended that other travel vehicles, such as bicycles, should be promoted to reduce vehicular emissions in mountainous terrain.

## 5. Conclusions

The study has been conducted to evaluate the associated cancer risk due to environmental conditions in the Indian Himalayas. The findings of the present study reflected that Manali comprises three types of tourism potential regions, i.e., high, medium and low TP regions. Among the high TP regions are Manali city (i.e., the Mall Road area), the Rohtang Pass and Solang Valley. Therefore, the TCC of Manali city was evaluated, and it was found that the threshold for the TCC of Manali was 300,000 tourists/month. In 2017, the number of tourists visiting Manali was observed to surpass this threshold in the months of April, May and June. During these months, fossil fuels-powered vehicles were also used to support the tourism industry and the contamination of different environmental media with carcinogenic compounds such as EC was also observed near the Hamta glacier. Contamination of the various environmental media with EC did not pose substantial carcinogenic health impacts on adults. However, children were reported to be more susceptible to health risks and a further increase in the atmospheric concentration of the EC in the IWHs can prove harmful for the regional community. There is a need to take corrective measures for the sustainable and controlled development of tourism in this region.

## 6. Limitations of the Study

During the field visit, only ten samples of water and air were collected, which are generally considered to be insufficient to estimate the carcinogenic health impacts. However, considering the harsh mountainous locations of the Himalayas, in this study the carcinogenic health impacts were estimated using these limited number of samples. It is suggested that future field visits be conducted to collect the larger number of samples.In this study, only the environmental samples were collected to estimate the impact of EC on the human health. It would have been more relatable if a certain number of human blood samples could be collected and analyzed.Seasonal variation in the concentration of EC in different environmental media could also be estimated.

## Figures and Tables

**Figure 1 ijerph-19-06459-f001:**
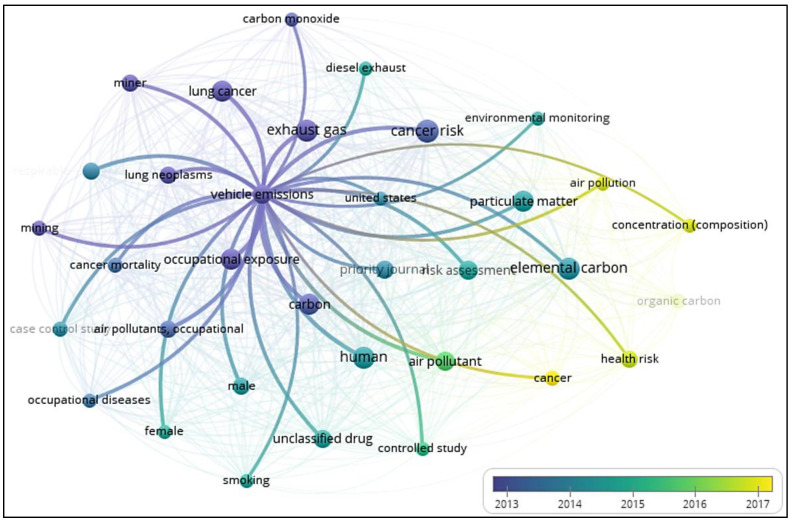
Keyword analysis of recent studies evaluated through Vosviewer.

**Figure 2 ijerph-19-06459-f002:**
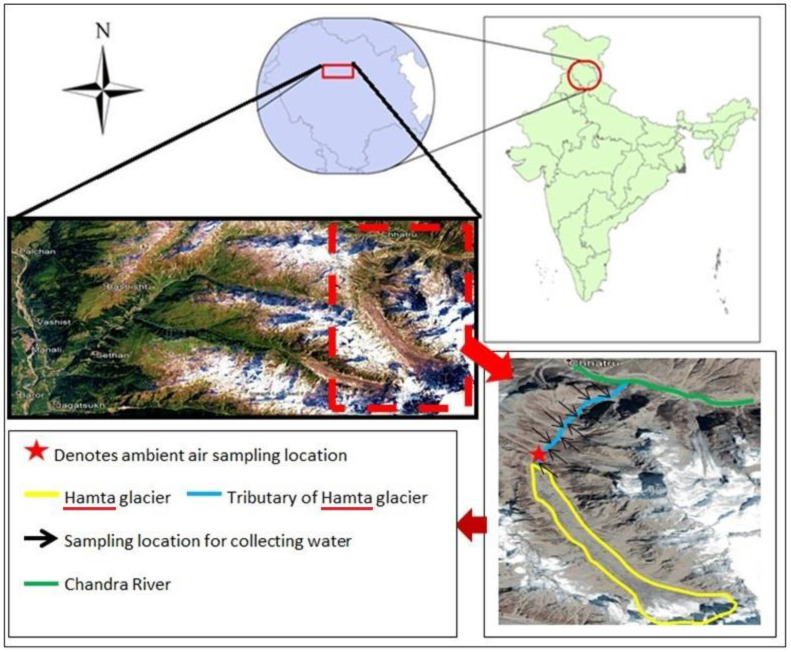
Geographic location of the site for collecting the water samples.

**Figure 3 ijerph-19-06459-f003:**
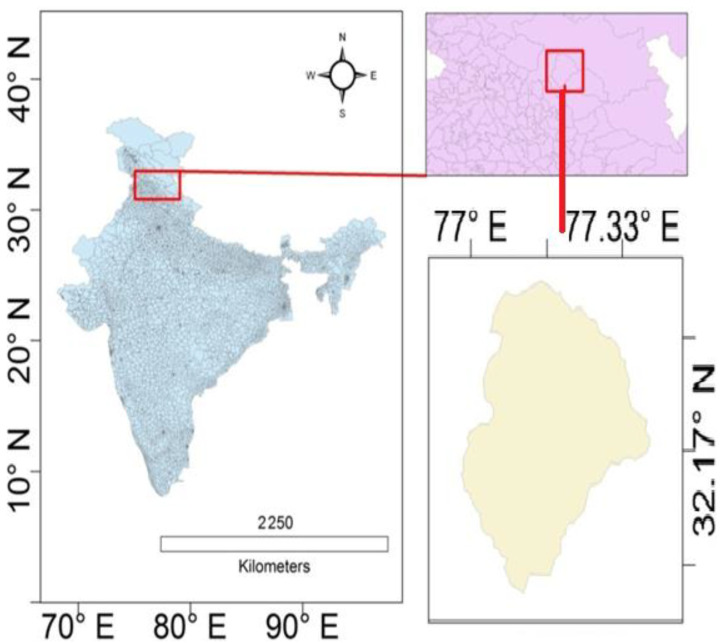
Geographical location of Manali, Himachal Pradesh, India.

**Figure 4 ijerph-19-06459-f004:**
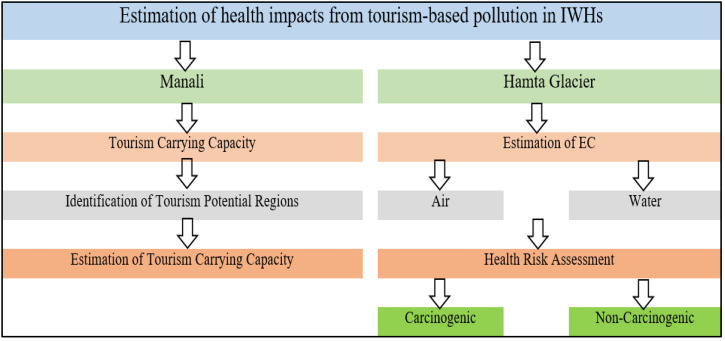
Brief methodology adopted during the study.

**Figure 5 ijerph-19-06459-f005:**
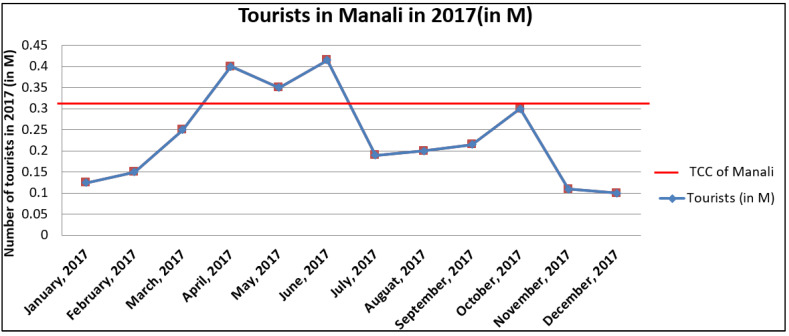
Comparison of monthly tourism (2017) with TCC in Manali.

**Table 1 ijerph-19-06459-t001:** Operating conditions for AE51.

Operating Parameters	Units	Operating Conditions AE 51
Wavelength number	-	Single-wavelength
Volumetric flow rate	m^3^ s^−1^	1.07 × 10^−6^
Measurement period	S	1, 60, 300
Sample spot area	m^2^	7 × 10^−6^
Deposition velocity	m s^−1^	0.26
Attenuation parameter	m^2^ g^−1^	11.5

**Table 2 ijerph-19-06459-t002:** Parametric evaluation of Tourism Potential regions.

Indicators	Weight	Normalized Weights for Indicators	Sub-Categories	Adjusted Rank	Normalized Weights for Sub-Categories
Natural Resources	4	0.25	Forests	6	0.24
Cultural Resources	2	0.12	Meadow	5	0.22
			Rivers	3	0.14
			Lakes	3	0.14
			Hot and cold water springs	4	0.11
			Glaciers	4	0.10
			Historical Monuments	4	0.6
			Informational art	1	0.5
			Cultural-organizations	2	0.4
Adventure Sports	3	0.25	Camping and trekking	3	0.4
			Ice-related sports	4	0.20
			Water sports	3	0.10
			Hunting and fishing	2	0.1
			Tourist accommodation	3	0.4
Facilities & Infrastructure	5	0.38	Transport and communication	3	0.3
			Infrastructure	4	0.20
			Shopping	3	0.2
			Gardens	1	0.2

**Table 3 ijerph-19-06459-t003:** Description of exposure parameters of ingestion and inhalation.

Denotations	Exposure Parameter (Units)	Standard Values	References
C	Concentration (Ingestion: mg kg^−1^) and (Inhalation: µg m^−3^)	-	-
BW	Body Weight (kg)	Adult = 75; Children = 15	[16]
EF	Exposure Frequency (d y^−^)	365 (for both adults and children)	[34]
ED	Exposure Duration (y)	Adult = 30; Children = 6	[35]
LT	Life Time (y)	65 (for both adults and children)	[33]
AT	Average exposure Time (y)	For Non-carcinogenic: ED × 365; For carcinogenic: LT × 365	[34]
IR_i_	Rate of ingestion (L d^−^)	2 for adults and 0.7 for children	[35]
IR_j_	Rate of inhalation( $m^{3}$ d^−1^)	20 for adults and 7.6 for children	[17,29,38]

**Table 4 ijerph-19-06459-t004:** Slope factor (mg kg^−1^ d^−1^)^−1^ of benz[a]pyrene for different exposure pathways.

S.No.	Exposure Pathway	Slope Factor	Reference
1	Ingestion	11.5	Masters (2000)
2	Inhalation	6.11	Masters (2000)
3	Dermal	2.5	Masters (2000)

**Table 5 ijerph-19-06459-t005:** Input values for different parameters used in the estimation of DA and SF.

S.No.	Parameters	Descriptions	Values
1	K	Coefficient of permeability of the skin (cm h^−1^)	0.001
2	t	Time of contact with skin (h d^−1^)	0.4
3	CF	Conversion factor	0.001
4	H	Height of an adult and children (cm)	165 and 114, respectively

**Table 6 ijerph-19-06459-t006:** Estimation of Tourism Carrying Capacity of Manali.

Denotation	Parameters’ Description	Manali City
A	Area for recreation	30 × 10^6^ m^2^
A_u_	Minimum area required by a tourist	15 m^2^
R_f_	Rotation factor	1
PCC	Physical carrying capacity	2 × 10^6^
cf_1_	Temperature limiting factor	0.5
cf_2_	Rainfall limiting factor	0.65
cf_3_	Infrastructure limiting factor	0.9
cf_4_	Transport limiting factor	0.88
cf_5_	Management limiting factor	0.65
cf_6_	Perception limiting factor	0.91
TML	Total magnitude of limiting factor	1.52
TCC	Tourism carrying capacity	0.3 M

**Table 7 ijerph-19-06459-t007:** Concentration of EC in different environmental media.

Sample No.	Concentration in Water Samples (ppb)	Concentration in Air Samples (µg/m^3^)
1	19.02	870
2	48.14	922
3	46.55	850
4	48.79	860
5	49.04	922
6	54.00	950
7	51.97	840
8	20.37	840
9	35.20	850
10	44	940

**Table 8 ijerph-19-06459-t008:** Life time cancer risk assessment (×10^−6^).

Glaciers		Life Time Cancer Risk					
		Ingestion		Inhalation		Dermal	
	Year	Adults	Children	Adults	Children	Adults	Children
Hamta	2017	0.04	0.16	0.02	0.49	0	0.05

## Data Availability

The data presented in this study are available on request from the corresponding author.

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
