# Peer review of "Impact of Unsustainable Environmental Conditions Due to Vehicular Emissions on Associated Lifetime Cancer Risk in India: A Novel Approach"

_ijerph, 2022, doi:10.3390/ijerph19116459_

Round 1

Reviewer 1 Report

Impact of unsustainable environmental conditions due to vehicular emissions on associated lifetime cancer risk in India: A novel approach

 This manuscript reports the study focused on estimating the tourism carrying capacity (TCC) of one of the world-famous tourist spot of the Indian Western Himalayas (IWHs) i.e., Manali, Himachal Pradesh, India.

The article presents valuable data of samples obtained in different environmental media i.e., water and ambient air, collected near the snout of the Hamta glacier. Life time carcinogenic and non-carcinogenic risk assessment of the residents exposed to contaminated environment, of the study area, was also evaluated.

Given its characteristics, resistance to atmospheric degradation, long range atmospheric transport, small size, capacity of disturb heart and autonomic nervous system, EC is highly toxic and very hazardous to health. In this article a quantification of the health risk to the EC is made.

The article is well written, and the results are consistent with the methodologies adopted.

As conclusion, the authors consider that contamination of the various environmental media with EC, did not pose substantial carcinogenic health impacts on adults.

I suggest that the authors make a comparison with other touristic places in India. I also suggest that the authors present some data on the incidence of cancer in the regions studied.

In my opinion Figures 2 and 3 have poor quality.

Author Response

Responses to the reviewers’ comments

Reviewer #1

General: This manuscript reports the study focused on estimating the tourism carrying capacity (TCC) of one of the world-famous tourist spot of the Indian Western Himalayas (IWHs) i.e., Manali, Himachal Pradesh, India.

The article presents valuable data of samples obtained in different environmental media i.e., water and ambient air, collected near the snout of the Hamta glacier. Life time carcinogenic and non-carcinogenic risk assessment of the residents exposed to contaminated environment, of the study area, was also evaluated.

Given its characteristics, resistance to atmospheric degradation, long range atmospheric transport, small size, capacity of disturb heart and autonomic nervous system, EC is highly toxic and very hazardous to health. In this article a quantification of the health risk to the EC is made.

The article is well written, and the results are consistent with the methodologies adopted.

Response: The authors are grateful to the reviewer for appreciating the efforts and acknowledging the importance of the findings of this study. Thank you.

Comment #1: As conclusion, the authors consider that contamination of the various environmental media with EC, did not pose substantial carcinogenic health impacts on adults.

I suggest that the authors make a comparison with other touristic places in India. I also suggest that the authors present some data on the incidence of cancer in the regions studied.

Response: Kindly note that the authors have made significant efforts to collect the suggested data, however, unfortunately, they were not able to meet the success. Therefore, it is requested to the reviewer regarding to skip this suggestion. Thank you for your valuable input.

Comment #2: In my opinion Figures 2 and 3 have poor quality.

Response: The authors have tried to improve the quality of the figures and shared them in the revised manuscript. Thank you.

Authors have carefully considered the comments from the editor and reviewers. Also, they have carefully dealt with the issues raised by editors and reviewers, so as to keep up with the quality of the IJERPH. We are now hopeful that this submission will be considered for publication in IJERPH.

Once again, thank you all, for your efforts in improving the quality of this manuscript.

Reviewer 2 Report

The research attempts to study the effects of tourists and tourism related activities with environmental conditions of Indian Western Himalayas. However, and to begin with, the figure obtained as "3,05,000" is incomprehensible. What is the meaning of "3,05,000" as the threshold of pollution? The same figure is repeated in different parts of the paper. After clearing which number is "3,05,000" the relation between this and the methods cannot be taken for granted. The authors assert that "according to the table 4, it was inferred that TCC for Manali is 3,05,000 tourists/month". That deserves further explanation as it's not obvious at all. The different analytical techniques currently used to measure EC do not show agreement for many PM samples. Studies that use EC as a tracer and integrate different analytical techniques for EC can significantly bias estimates of source contributions to atmospheric PM. In addition, source attribution studies that do not properly address all sources of EC in the atmosphere can also lead to inaccuracies and biases. Sentences such as "Flourishing tourism industry of Manali, demands fossil fuels'-driven vehicles which acts as a potential source of EC" make no sense. Similarly, the expression "lakh" can be meaningful in India, but not beyond.

The equations are very hard to follow with the abuse of acronyms. The complete noun is needed as well as the relations among them. The result is not in line with the equations. Asserting that "life time cancer risk from EC contaminated atmosphere was estimated for adults and children and results didn't depict any significant impacts with the current rate of EC emissions, however, carcinogenic impacts in future can be presumed" is not a logical inference from the methods and the result should be more explained. Which are the levels of EC emissions that impair health? What can be presumed in future and why/why not? 

Author Response

Responses to the reviewers’ comments

Reviewer #2

Comment #1: The research attempts to study the effects of tourists and tourism related activities with environmental conditions of Indian Western Himalayas. However, and to begin with, the figure obtained as "3,05,000" is incomprehensible. What is the meaning of "3,05,000" as the threshold of pollution? The same figure is repeated in different parts of the paper. After clearing which number is "3,05,000" the relation between this and the methods cannot be taken for granted. The authors assert that "according to the table 4, it was inferred that TCC for Manali is 3,05,000 tourists/month". That deserves further explanation as it's not obvious at all.

Response: The threshold value of 0.3 M was finalized using TCC calculation and then the number of tourists visiting Manali per month were compared with it.

Figure 8 (in the revised manuscript) illustrated that the number of tourists visiting Manali is below the ERCC throughout the year except for the few months.  Further, it can also be see that during the months of April, May, June and October the flow of tourists surpassed the TCC of Manali. During these months’ various urban management problems such as traffic jams, unavailability of rooms, littering of solid waste etc., are faced by the local authorities. It is hoped that this explanation is appropriate to explain the setting of 0.3 M as the threshold value for inflow of the tourists in Manali.

Moreover, international SI units are incorporated in the revised manuscript for better understanding. Thank you.

Comment #2: The different analytical techniques currently used to measure EC do not show agreement for many PM samples. Studies that use EC as a tracer and integrate different analytical techniques for EC can significantly bias estimates of source contributions to atmospheric PM.

Response: The authors agree with the reviewer however, the different analytical techniques for estimation of EC were adopted from available literature. Some of the studies that used similar techniques are as follows:

  1. Wang M, Xu B, Zhao H, Cao J, Joswiak D, Wu G, Lin S (2012) The influence of dust on quantitative measurements of black carbon in ice and snow when using a thermal optical method. Journal of Aerosol Science and Technology 46: 60–69.
  2. Thind PS, Chandel KK, Kumar S, Mandal TK, John S (2019) Enhanced snow melting due to the deposition of Light Absorbing Particulates in the Indian Western Himalayas. Environmental Science and Pollution Research doi.org/10.1007/s11356-019-04183-5.
  3. Doherty SJ., Grenfell TC, Forsström S, Hegg DL, Warren SG, Brandt R (2012) Observed vertical redistribution of black carbon and other light-absorbing particles in melting snow. Journal of Geophysical Research- Atmospheres, 118, 5553-5569. doi:10.1029 /2012JD018956.

Therefore, the methodology adopted in this study was based on previous published studies and hence, can be verified. Thank you.

Comment #3: In addition, source attribution studies do not properly address all sources of EC in the atmosphere can also lead to inaccuracies and biases.

Response: In the current study, the sources of EC were presumed from a study conducted by Thind et al. (2021) i.e.:

“Thind, P.S., Kumar, D., and John, S., 2021. Source apportionment of the light-absorbing impurities present in surface snow of the India Western Himalayan glaciers. Atmospheric Environment, 246, p.118173”.

This study made it evident that the major sources of EC in the IWHs are vehicular pollution and biomass burning. Contribution from industrial burning was also reported and was found to be insignificant. Therefore, in this study, the authors also considered emissions from vehicular pollution, owing to tourism, as the major source of EC. Thank you.

Comment #4: Sentences such as "Flourishing tourism industry of Manali, demands fossil fuels'-driven vehicles which acts as a potential source of EC" make no sense.

Similarly, the expression "lakh" can be meaningful in India, but not beyond.

Response: Kindly note that the corrections have been made in the revised manuscript and SI units have been used. Thank you.

Comment #5: The equations are very hard to follow with the abuse of acronyms. The complete noun is needed as well as the relations among them. The result is not in line with the equations.

Response: Authors agree with the reviewer as they have incorrectly mentioned the ERCC and TCC in the previous version of the manuscript. However, in the revised manuscript this error has been eliminated. Thank you for identifying the mistake.

Comment #6: Asserting that "life time cancer risk from EC contaminated atmosphere was estimated for adults and children and results didn't depict any significant impacts with the current rate of EC emissions, however, carcinogenic impacts in future can be presumed" is not a logical inference from the methods and the result should be more explained. Which are the levels of EC emissions that impair health? What can be presumed in future and why/why not?

Response: It should be noted that when the value of lifetime cancer risk is >10-6, the carcinogenic impacts can be speculated. However, when the value is closer or between 10-4 to 10-6, the non-carcinogenic health impacts may take place. Therefore, with the increasing air pollution and emission of EC, the value of lifetime cancer risk can be presumed to increase. In this way, the EC-contaminated environment may impact human health in mountainous regions. A similar explanation has also been added in the revised manuscript. Thank you.

Authors have carefully considered the comments from the editor and reviewers. Also, they have carefully dealt with the issues raised by editors and reviewers, so as to keep up with the quality of the IJERPH. We are now hopeful that this submission will be considered for publication in IJERPH.

Once again, thank you all, for your efforts in improving the quality of this manuscript.

Reviewer 3 Report

Remarks:

  •  the used methods were not suitable.
  • the results were presented but without deeper analysis
  • the conclusions were not supported by the data.
  • the number of samples is so small that it does not guarantee representative results
  • Fig. 2 to 6 are redundant
  • Authors do not use international numbering system
  • References section needs improvement (outdated)

Author Response

Responses to the reviewers’ comments

Reviewer #3

Remarks:

Comment #1: The used methods were not suitable.

Response: The authors agree with the reviewer however, the different analytical techniques for estimation of EC were adopted from available literature. Also, in the revised manuscript a brief chart of methodology has also been added for better understanding of the readers. Some of the studies that used similar techniques are as follows:

  1. Wang M, Xu B, Zhao H, Cao J, Joswiak D, Wu G, Lin S (2012) The influence of dust on quantitative measurements of black carbon in ice and snow when using a thermal optical method. Journal of Aerosol Science and Technology 46: 60–69.
  2. Thind PS, Chandel KK, Kumar S, Mandal TK, John S (2019) Enhanced snow melting due to the deposition of Light Absorbing Particulates in the Indian Western Himalayas. Environmental Science and Pollution Research doi.org/10.1007/s11356-019-04183-5.
  3. Doherty SJ., Grenfell TC, Forsström S, Hegg DL, Warren SG, Brandt R (2012) Observed vertical redistribution of black carbon and other light-absorbing particles in melting snow. Journal of Geophysical Research- Atmospheres, 118, 5553-5569. doi:10.1029 /2012JD018956.

Therefore, the methodology adopted in this study was based on previous published studies and hence, can be verified. Thank you.

Comment #2: The results were presented but without deeper analysis

Response: An elaborated discussion has been added in the revised manuscript. Thank you for your valuable suggestions.

Comment #3: The conclusions were not supported by the data.

Response: References to background and supporting data have been added in the revised manuscript. Thank you.

Comment #4: The number of samples is so small that it does not guarantee representative results.

Response: In current study, the water samples were collected from stream of Hamta glacier which was just of length 1 km. Considering the length of stream and area it covers, the number of samples i.e., 10, can be considered. Moreover, as the sample location is situated at harsh conditions of the Himalayas, the logistic support was limited. Therefore, the authors were able to collect only 10 samples. Considering these two arguments, it is requested to kindly accept the number of samples for current study. Thank you.

Comment #5: Fig. 2 to 6 are redundant

Response: The quality of figures has been improved and new figures have been added in the revised manuscript. Thank you.

Comment #6: Authors do not use international numbering system

Response: Only SI units have been used in the revised manuscript. Thank you.

Comment #7: References section needs improvement (outdated)

Response: Kindly note that as suggested by the reviewer, the authors have also added latest and updated references in the revised manuscript. Thank you.

Authors have carefully considered the comments from the editor and reviewers. Also, they have carefully dealt with the issues raised by editors and reviewers, so as to keep up with the quality of the IJERPH. We are now hopeful that this submission will be considered for publication in IJERPH.

Once again, thank you all, for your efforts in improving the quality of this manuscript.

Reviewer 4 Report

As a reviewer I have the following remarks. I am using “you” as Dear Authors.

  1. Line 81: “(and Raatikainen et” – something missing before “and”.
  2. Line 85: “fossil fuels combustions” – do you mean here deforestation?
  3. Line 109: “to encourage eco-tourism” vs. Line 97: “due to uncontrolled inflow of tourists” – could you please provide some differences between these types
  4. Line 100: Figure 1 – could you please specify what we see this picture.
  5. Line 113: “10 water samples were collected from the stream” – in the same site?
  6. Line 121: “in Figure3” – Figure 3.
  7. Equation 6. What was value for n in this study?
  8. Line 256: “carcinogenic impacts in future can be presumed” – do we have an effect of the accumulation? Say, even pollutants 100 years old, also transboundary.
  9. In this work so much is indicated on tourism as a main source of pollution. I think there are others as well?

Thank you

Author Response

Responses to the reviewers’ comments

Reviewer #4

As a reviewer, I have the following remarks. I am using “you” as Dear Authors.

Comment #1: Line 81: “(and Raatikainen et” – something missing before “and”.

Response: Kindly note that typo has been removed from the revised manuscript. Thank you for your efforts.

Comment #2: Line 85: “fossil fuels combustions” – do you mean here deforestation?

Response: In this statement by mentioning “fossil fuels’ combustion” the authors wanted to highlight the emission of pollutants by burning of diesel/petrol in vehicles. Thank you.

Comment #3: Line 109: “to encourage eco-tourism” vs. Line 97: “due to uncontrolled inflow of tourists” – could you please provide some differences between these types

Response: It is worth noting that the term eco-tourism defines the sustainable tourism in which the incoming tourism does not impact the natural resources of the place significantly. In case of uncontrolled inflow of tourists, without the implementation of eco-tourism, several environmental impacts can be speculated such as, waste generation, accumulation, forest fires, biomass burning etc. Therefore, this study highlights the importance of eco-tourism in Manali, India. Thank you.

Comment #4: Line 100: Figure 1 – could you please specify what we see in this picture.

Response: As VOSviewer is a platform for developing and envisioning peer-reviewed citation-based bibliometric networks. Such connectivity frameworks can indeed be created by utilizing citations, scholarly metadata-based-bibliometric coupling, co-citation, as well as co-authorship connections, which can include publications, scientists/investigators/scholarly experts/professionals, and perhaps independent publications. In other terms, bibliographical-couplings refer to the convergence in published relevant identified studies and reference sources. VOSviewer's characteristic is notably valuable for portraying massive scientometric mapping charts in an obvious convenient format.

Comment #5: Line 113: “10 water samples were collected from the stream” – in the same site?

Response: The water samples were collected from a single stream however, at different locations. The location of these samples is shown by black-colored arrows in the Figure 2.

Comment #6: Line 121: “in Figure3” – Figure 3.

Response: Typo corrected. Thank you.

Comment #7: Equation 6. What was value for n in this study?

Response: As can also be seen from Table 6, the value of ‘n’ is 6. Thank you.

Comment #8: Line 256: “carcinogenic impacts in future can be presumed” – do we have an effect of the accumulation? Say, even pollutants 100 years old, also transboundary.

Response: This statement presumes that the trend of emission of EC will keep on increasing with time. Therefore, in coming years, when the concentration of EC increases beyond certain limit it may cause carcinogenic impacts on human beings. Thank you.

Comment #9: In this work so much is indicated on tourism as a main source of pollution. I think there are others as well?

Response: In a recent study, Thind et al. (2021) i.e.:

Thind, P.S., Kumar, D. and John, S., 2021. Source apportionment of the light absorbing impurities present in surface snow of the India Western Himalayan glaciers. Atmospheric Environment, 246, p.118173.

made it evident that the major sources of EC in the IWHs are vehicular pollution and biomass burning. Contribution from industrial burning was also reported and was found to be insignificant. Therefore, in this study, the authors also considered emissions from vehicular pollution, owing to tourism, as the major source of EC. Thank you.

Authors have carefully considered the comments from the editor and reviewers. Also, they have carefully dealt with the issues raised by editors and reviewers, so as to keep up with the quality of the IJERPH. We are now hopeful that this submission will be considered for publication in IJERPH.

Once again, thank you all, for your efforts in improving the quality of this manuscript.

Round 2

Reviewer 3 Report

Dear Authors,

I appreciate the changes that have been introduced in the text of the work. However, I maintain my earlier opinion that methodological errors were made during the measurements and the obtained results are not representative and cannot be the basis for the health exposure assessment. Figures 1, 3, 4, 5, 6 and 7 are redundant - they do not add any information to the presented content. There is no explanation for the relationship between TCC and water and air pollution at the test site. Since the stream from which the water was taken is only 1 km long, how many inhabitants use it? Determination of EC / water and air B(a)P content should be performed for the carcinogen exposure assessment. The EC concentrations of the samples are not given, but only the range. So what values were inserted into the oral and dermal intake formulas? No inhalation intake has been assessed. 

Author Response

Responses to the reviewers’ comments

Reviewer

General: I appreciate the changes that have been introduced in the text of the work. However, I maintain my earlier opinion that methodological errors were made during the measurements and the obtained results are not representative and cannot be the basis for the health exposure assessment.

Response: Authors would like to thank the reviewer for appreciating the revisions made to the manuscript. In this version, the authors have made substantial efforts to eliminate the methodological errors and meet the expectations of the reviewer.

Comment #1: Figures 1, 3, 4, 5, 6, and 7 are redundant - they do not add any information to the presented content.

Response: As suggested by the reviewer, the authors have removed Figures 5, 6, and 7 from the revised manuscript. However, it is suggested to keep the other figures and justification for that each as follows:

Figure 1: This figure presents the results of the literature review on different aspects of air pollution and its consequences with respect to environmental health. This Figure assists in establishing the areas where this study is unique and hence, adds novelty to the present study.

Figure 3: This figure presents the geographical location of Manali, Himachal Pradesh, India. This figure will assist the international readers in easily identifying the location of the study area. In this way, they will feel more connected to the study, and coherence is maintained.

Figure 4: This figure defines the overall methodology of the present study which, makes it easier for the readers to understand. Therefore, it is requested to keep this figure.

It is hoped that the reviewer will agree with the authors are agree to keep these figures. Thank you for your valuable suggestions.

Comment #2: There is no explanation for the relationship between TCC and water and air pollution at the test site.

Response: Due to the ongoing COVID-19 situation resulting in limited access to testing facilities, we have tried our best to evaluate the best possible relationships by examining the TCC, water, and air pollution. However, in the future, we will do the recommended analyses. Kindly note that the authors have made significant efforts to collect the suggested data, nevertheless, unfortunately, they were not able to meet the success. Therefore, it is requested to the reviewer to kindly please skip this suggestion. Thank you for your valuable input.

Comment #3: Since the stream from which the water was taken is only 1 km long, how many inhabitants use it?

Response: Authors are thankful to the reviewer for giving them the opportunity to explain this matter. In the Himalayan region of India, the major source of fresh water for a large portion of the inhabitants is snow-melted water.

The reference article supporting this argument is entitled as, “Immerzeel, W.W., Van Beek, L.P. and Bierkens, M.F., 2010. Climate change will affect the Asian water towers. science, 328(5984), pp.1382-1385”.

There are other similar studies which can also strengthen this argument. Therefore, in this study, contamination of snow-melted water was assessed. Thank you.

Comment #4: Determination of EC / water and air B(a)P content should be performed for the carcinogen exposure assessment.

Response: In the present study, the contamination of water and air samples with EC was estimated, near Hamta glacier. Further, the carcinogenic exposure was assessed for the same. However, in the case of air exposure i.e., inhalation, the slope factor of EC was considered equivalent to B[a]P following the findings of another study which is entitled as, “Ravindra, K (2019) Emission of black carbon from rural household’s kitchens and assessment of lifetime excess cancer risk in villages of North India. Environment International 122: 201-212”. In this way, the carcinogenic health impacts of EC on humans were assessed. Thank you.

Comment #5: The EC concentrations of the samples are not given, but only the range. So what values were inserted into the oral and dermal intake formulas?

Response: Kindly find the concentration of EC (ppb) in ten water samples as follows:

  1. 02
  2. 14
  3. 55
  4. 79
  5. 04
  6. 00
  7. 97
  8. 37
  9. 20
  10. 00

Thank you.

Comment #6: No inhalation intake has been assessed.

Response: Kindly note that section 3.4.1. entitled, “Health risk through oral intake” mentions the oral intake that defines the ingestion and inhalation of EC. Thank you.

The authors have carefully considered the comments from the editor and reviewers. Also, they have carefully dealt with the issues raised by editors and reviewers, so as to keep up with the quality of the IJERPH. We are now hopeful that this submission will be considered for publication in IJERPH.

Once again, thank you all, for your efforts in improving the quality of this manuscript.

Note: All the necessary changes/added sentence has been shown in yellow font.

Thank you very much in advance for taking your time in reviewing this manuscript.

Sincerely, we hope you will find our revision satisfactory.

Thanks, in anticipation.

Warm regards,

Shubham Sharma

Gianpaolo Di Bona

(Corresponding authors)
